# Neuron-Specific Enolase (NSE) as a Biomarker for Autistic Spectrum Disease (ASD)

**DOI:** 10.3390/life13081736

**Published:** 2023-08-13

**Authors:** Felician Stancioiu, Raluca Bogdan, Radu Dumitrescu

**Affiliations:** 1Fundatia Bio-Forum, 040245 Bucharest, Romania; felicians@bio-forum.net; 2Medicover Hospital Bucharest, 013982 Bucharest, Romania

**Keywords:** autistic spectrum disease (ASD), autism, neuron-specific enolase (NSE), autism biomarker, neuroinflammation, neuronal apoptosis, mTOR

## Abstract

Autistic spectrum disease (ASD) is an increasingly common diagnosis nowadays with a prevalence of 1–2% in most countries. Its complex causality—a combination of genetic, immune, metabolic, and environmental factors—is translated into pleiomorphic developmental disorders of various severity, which have two main aspects in common: repetitive, restrictive behaviors and difficulties in social interaction varying from awkward habits and verbalization to a complete lack of interest for the outside world. The wide variety of ASD causes also makes it very difficult to find a common denominator—a disease biomarker and medication—and currently, there is no commonly used diagnostic and therapeutic strategy besides clinical evaluation and psychotherapy. In the CORDUS clinical study, we have administered autologous cord blood to ASD kids who had little or no improvement after other treatments and searched for a biomarker which could help predict the degree of improvement in each patient. We have found that the neuron-specific enolase (NSE) was elevated above the normal clinical range (less than 16.3 ng/mL) in the vast majority of ASD kids tested in our study (40 of 41, or 97.5%). This finding opens up a new direction for diagnostic confirmation, dynamic evaluation, and therapeutic intervention for ASD kids.

## 1. Introduction

Since its first description in a series of 11 children by Leo Kanner 80 years ago [1], ASD is being more frequently diagnosed, but in doing so, its estimated direct and indirect economic costs in the USA will likely reach the 500 billion USD mark in 2025, making this pathology more costly than diabetes [2].

The mainstay of ASD treatment is psychotherapy, which typically involves a long and intensive schedule in order to help kids make significant progress; additionally, neurofeedback; antipsychotics, such as risperidone and aripiprazole; stimulants, such as atomoxetine; and other drugs, such as oxytocin and bumetanide, are prescribed infrequently. More recently, stem cell therapies (both hematopoietic and mesenchymal) and exosomes were reported to produce improvements in certain kids [3,4,5,6,7]; however, ASD has many aspects yet unknown, including pathophysiology and treatment, which makes its management challenging.

Abnormal findings related to ASD were reported even before birth, with identification of maternal antibodies to fetal brain protein in mid-pregnancy [8,9]. Immune dysfunction seems to be a pervasive aspect in ASD, identified in many studies [10,11,12,13], including post-mortem [14], as the increased levels of various inflammatory cytokines such as IL-1β, IL-6, IL-8, TNF-α, the macrophage chemoattractant protein (MCP)-1, the transforming growth factor (TGF)–β1, etc. [15,16,17]. Increased GFAP and NFkB specifically underscores neuroinflammation and glial M1 activation in the Central Nervous System (CNS) [18,19,20,21]; recently, we have more complete and precise profiles of the inflammation present in ASD [22].

Electrophysiological testing such as resting-state electroencephalography (EEG), power spectral densities, and the Fitting Oscillations and One-Over-F algorithm can show the differences between ASD and typically developing (TD) children [23]. Dynamic EEG testing involving face perception, visual evoked potentials, and the perception of biological motion as specific findings in ASD [24], and successful therapeutic intervention reflected in better social interaction was associated with increased alpha and beta power, and decreased theta power [6].

The CNS modifications in ASD are more difficult to define with neuroimaging because on top of the clinical heterogeneity of ASD are added variations in standardization of imaging protocols [6,25]; however, specific features were found in ASD, such as functional connectivity, gray matter volume, and differences of cortical volume in certain areas [26,27]. Differences were also found in the growth pattern with age [28], in the enlarged amygdala and hippocampus in childhood, but not in adolescence [29,30,31]. The largest imagistic study to date—the ENIGMA ASD study [32]—included a total of 1571 ASD patients and 1651 TD controls from 13 countries and found both similar and opposite modifications with other studies: ASD patients had smaller putamen, pallidum, amygdala, nucleus accumbens, larger lateral ventricles, and mean intracranial volume.

The genetic picture in ASD is also complex given the number of genes involved—about 800 genes with their respective myriad combinations—and even though only 20–50% of ASD cases can be clearly linked to genetic variants [33,34,35], there is a multitude of possible gene alterations: copy number variations, single nucleotide polymorphism (SNPs), inversions, deletions, insertions, duplications, the epigenetic control of transcriptions such as modifications of promoter regions [36,37], the unrelated SNPs converging to alter cellular pathways [38], and finally the diverse phenotypic consequences of genetic alterations. All of these factors mentioned above highlight the difficulties of finding a common denominator in ASD pathology, which can be reflected in a universal biomarker.

Adding to the complexity of the genetic determinants of ASD is the influence of environmental factors on vulnerable genetic material, factors which may alter the transcription and translation of the genetic material directly via interaction with the DNA, or indirectly through impairing the DNA repair or oxidative damage [39]. Some examples of the common types of genes affected are those involved in synapse development—SH3, NLGN, SHANK, CNTNAP2, NRXN, and pathways controlling chromatin remodeling and transcriptional modulation—FMRP, RBFOX, MeCP2, ADNP, etc. [40]. Similar to imagistic studies, genetic studies have revealed age-related differences in ASD [41].

Besides sequencing and microarrays, another way of investigating the genetic modifications in ASD is searching for specific plasma or cerebrospinal fluid nucleic acids: circulating, non-coding RNAs (micro RNAs—miRs—and long non-coding RNAs—lncRNAs); these control gene transcriptions involved in cell signaling, cell adhesion, metabolism, and also cancer [42], and miR-181b-5p, miR-320a, miR-572, miR-130a-3p, and miR-19b-3p—are specific to the ASD pathology [43]. While some micro RNAs—miR-181b-5p and miR-328—were downregulated in both the brain and peripheral blood, miR-106b-5p was shown to be upregulated in the brain of ASD patients in a post-mortem study [44] and downregulated in the peripheral blood (lymphoblastoid lines). A different miRNA profile for ASD was observed by another team [45] and involved miR-197- 5p, miR-3135a, miR-328-3p, miR- 365a-3p, miR-424-5p, miR-500a-5p, miR-619-5p, and miR-664a-3p; these miRNAs targeted the voltage-dependent calcium channel subunits alpha-1C and beta-1 (CACNA1C and CACNB1 genes) of the epigenetic controller gene DICER which codes the type III ribonuclease. Both authors prompt for validation of these markers in large patient cohorts.

The difficulty of the ASD biomarker conundrum is underscored by a recent meta-analysis [46] which evaluated 780 studies involving 120,000 ASD patients and 176,000 controls while testing various biomarkers: biochemical (374), neuroimaging (203), neurophysiological (133), neuropsychological (65), and 5 genetic (GWAS), yet an ASD biomarker with at least 80% sensitivity and specificity was still not found. Similar results were obtained via another recent meta-analysis [47] in which 940 biomarkers were analyzed; the least frequent were neuroimaging and neurophysiological biomarkers, and the most frequently studied were molecules such as cytokines, growth factors, oxidative stress indicators, neurotransmitters, and hormones. This analysis also showed the low reproducibility of the results of the biomarker studies. In this context, it is worthwhile mentioning that even the gold-standard of ASD diagnosis, the Autism Diagnostic Observation Schedule (ADOS), was shown to have a pooled sensitivity of 91%, a specificity of 73%, and a 52% overall diagnostic accuracy in a meta-analysis involving more than 4000 children [48].

More recent studies integrated the data from more than one type of investigation, such the electrophysiological and neurotransmitter data on excitatory/inhibitory balance (E/I on resting EEG), GABA and glutamatergic neurotransmitters, imagistics on cortical thickness, and gene expression from various databases, [49,50,51,52]. These findings are adding more pieces to the ASD puzzle. When testing a panel of biomarkers, the results show much better specificity and sensitivity compared to the single biomarkers, so that a panel of 12 inflammatory and immune biomarkers identified 87% of ASD cases [22], and an 83.3% sensitivity and 84.6% specificity was attained when proteomics testing and a panel of nine proteins was employed [53].

We also used a panel of blood tests while evaluating ASD children for the CORDUS clinical study in our search for a marker which could predict the efficacy of the autologous cord blood treatment administered during this study. Various markers of inflammation, metabolic, and neuroendocrine function were performed by an independent clinical laboratory, and, as a striking new finding, we have observed that most children with ASD had persistently elevated plasma values of the neuron-specific enolase (NSE) above the normal clinical range of 16.3 ng/mL.

An increased neuronal apoptosis with altered levels of Bcl-2 and p53 in the frontal and cerebellar cortex was observed in a postmortem study on ASD [54], and another study has showed the presence of specific NSE epitopes in the plasma of mothers of ASD kids [55] but testing for increased NSE values as a reflection of increased apoptosis in ASD was not conducted so far.

The patients tested were screened for enrollment in this clinical study; prior to enrolling and testing, informed consent was obtained from the children’s parents. The CORDUS Clinical Study is registered on www.clinicaltrials.gov with NCT04007224, and the ethics approval was given by the National Bioethics Committee of the Romanian Medicine Agency ANM: IS/4/12.02.2020.

## 2. Materials and Methods

Patients previously diagnosed with ASD (F84.0 or F84.1 on ICD-11) by a pediatric psychiatrist or psychotherapist and who have undergone previous treatments including psychotherapy were screened for the CORDUS clinical study and tested between April 2022 and April 2023 with a battery of tests, which included NSE. The tests analyzed were administered as the initial set of tests for each child, after which the children enrolled were tested again two times; these latter results are not analyzed in this article. During the first set of tests, the following plasma and serum parameters were tested: complete blood count with differential (CBC); electrolytes (sodium, potassium, and calcium); liver function (AST and ALT); renal function (urea and creatinine); inflammatory panel: erythrocyte sedimentation rate (ESR) and serum protein electrophoresis; C-reactive protein ultrasensitive (CRP); tumor necrosis factor alpha (TNF-α); ferritin; and also a panel of nine IgG antineuronal antibodies: Anti Amphiphysin, Anti CV2, Anti PNMA2 (Ma2/Ta), Anti Ri, Anti Yo, Anti Hu, Anti Recoverin, Anti SOX1, and Anti Titin.

In addition, IL-1 beta, IL-6, IL-8, IL-10, neopterin, procalcitonin, human growth hormones (hGH), thyroid stimulating hormones (TSH), the insulin-like growth factor-1 (IGF-1), homocysteine, IgE levels, the cationic protein of eosinophils, and S100 were performed sporadically in specific patients based on the clinical evaluation of the study investigator.

Blood was drawn and tests were performed at a clinical laboratory—Bioclinica Laboratory Bucharest—where NSE was determined via the electrochemiluminescence (ECLIA) method, with normal range being considered values of less than 16.3 ng/mL.

We have analyzed the test results in order to compare the frequency of abnormal values for the various blood markers and for some of these tests—including NSE—we have repeated the testing after a few weeks for confirmation of the elevated values.

Test results were entered in Microsoft Excel and descriptive statistics were performed also using this software.

## 3. Results

Blood test results were communicated by the independent clinical laboratory directly to patients (parents of children) and the study investigators; results are summarized in Table 1 and Table 2 below.

We have found that, besides TNF-α, the inflammatory cytokines which we have tested were not frequently modified; IL-6 and IL-1β was in the normal range for most of the kids tested (increased in 10% of kids tested), and there were similar results with the IL-8 testing. Only two kids tested positive for IgG antineuronal antibodies (one for anti-SOX-1 and one for anti-amphyphysin), and upon re-testing within 4 weeks from the initial testing, both kids were negative for the respective antibodies, suggesting a crossover reaction to a viral infection (the two kids had cold symptoms 1–3 weeks prior).

We have also considered a set of five blood markers and their abnormal values as positive inflammatory markers (IMs)—ESR, CRP, ferritin, α-2 globulins, and TNF-α. We have found that 51 of 55 patients had at least one of these IMs present; most patients (21) had two IMs, and four patients had no abnormal values on any of the five IMs (the data are summarized in Figure 1). All four patients with normal IMs had increased NSE values; two had an increased eosinophils percentage on CBC, two had slightly lower sodium levels (137 mmol/L), and one had microcytic anemia.

Finally, we have tested homocysteine levels in 30 kids, and of those, 4 had low homocysteine levels (13.3%) and 1 had an increased level (3.33%); 2 of 10 tested had low hGH (20%); another 2 kids from the 10 kids tested had low IGF-1, and 1 kid had high TSH; 5 children from 55 (9.09%) had increased MCV, most likely due to folate/vitamin B12 utilization deficit, and 1 of these children had documented antibodies to folate receptor (the results are summarized in Table 2).

One patient had increased bilirubin (both total and direct), and this was found by us after observing in this patient’s history an adverse reaction to risperidone (lethargic state), which prompted pharmacogenetic testing that showed SNPs in SLC15A2, SLC22A1, SLCO1B3, ABCB1, and ABCC2 genes (the latter is linked to Dubin–Johnson syndrome).

An electrolyte disturbance was observed in a few patients; we have ignored hyperpotassemia (10 instances) on the grounds of possibly hemolyzed blood specimens because of the difficult venipunctures in some ASD kids, but the observed lowered sodium is more likely linked to a functional alteration in the NKCC2, a cotransporter protein of Na^+^, K^+^, and Cl^−^ encoded via the SLC12A1 gene. This dysfunction is linked to GABA receptor activity and is improved via the administration of low-dose bumetanide—a loop diuretic—which was also shown to ameliorate symptoms of ASD [56]. More recently, torasemide, another Na^+^/K^+^ inhibitor, was proposed to be administered in ASD, with less side effects [57].

A relatively common occurrence among ASD children was an allergic-type reaction, with increased eosinophils observed in about 1/3 of the kids tested for this study (Table 2). While we have only seldom tested IgE levels and the cationic protein of eosinophils (and found them to be increased in the few kids tested), it is known that the increased levels of IgEs and the recurrent infectious episodes are associated with variants of the STAT3 genes and DOCK8 genes, with autosomal dominant transmission, which are shown below to be modified in ASD. STAT pathways can be downregulated via the administration of luteolin and diosmin. Additionally, quercetin inhibits the differentiation of Th1 triggered by neural antigens and IL-12 production [58].

## 4. Discussion

Elevated NSE values are known to be present in about 60–80% of patients with small cell lung carcinoma, neuroblastoma, seminoma, and, in some patients, with benign CNS tumors, neuroendocrine neoplasia—medullary thyroid carcinoma, pancreatic carcinoma, Merkel cell carcinoma, carcinoid tumors; however, there is no published data with the finding of increased NSE values in ASD.

Specific NSE epitopes were found in ASD kids [55]; however, in this study tests were performed from maternal (not the ASD child) plasma and this can confound the maternal vs. fetal origin of NSE. Also, since NSE is intracellular and released post-apoptosis, the primary pathological event is neuronal apoptosis and secondary the antibody production against the released NSE, which may also explain why a majority of kids have elevated NSE levels, but only around 23% show presence of autoantibodies.

Finally, the neuronal destruction in ASD is focal, not global—as shown by multiple anatomopathological, imagistic, and genetic studies [26,27,28,29,40,52,59,60]—and it is very likely that specific areas of the ASD child CNS (e.g.. prefrontal cortex, periventricular areas involved in neurogenesis, etc.) are affected by the simultaneous spatial and temporal convergence of genetic, immune, and metabolic factors. Therapeutic interventions to specifically stimulate these areas are currently limited either by the multiple and sometimes diffuse actions of various pharmacotherapies (non-selective receptor or neurotransmitter stimulation translated in adverse reactions, most frequently agitation) or by trivial, practical considerations —e.g., neurofeedback can be performedonly with the child’s cooperation, and many kids are too agitated to participate and achieve medium term results; however, a recent study showed that viloxazine has superior results to atomoxetine in ADHD kids [61] and gives hope for such a specific intervention in ASD. Viloxazine increases activity in the prefrontal cortex, possibly via differential actions on 5-HT receptors: antagonistic on 5-HT2B and agonistic on 5-HT2C receptors, and also acts on the norepinephrine transporter [62],—causing an important modulation of the cortico-basal tracts involved in the cortical control of the enlarged ASD amygdala [29,30,31], with improvement of the fear and anxiety symptoms which are present in a majority of ASD kids.

From a different perspective, the data from post-mortem studies and imagistic shows that in about 60% of ASD cases, there is an increase in synapses (due to excessive formation and/or lack of “pruning” of synapses) and neuron numbers, [59,63,64,65] which in most cases lead to non-functional networks; in exceptional cases, this zonal neuronal exuberance may explain the rare and very specialized abilities of kids formerly diagnosed with Asperger’s syndrome. Another observation is that NFkB is necessarily activated in both neuro-inflammation [19] and in synaptic formation [66], and besides underscoring the important role of inflammation in many ASD cases [67], it may mean that lowering neuro-inflammation without completely abolishing neuronal NFkB activation is a desirable therapeutic strategy which may reduce neuronal apoptosis and the NSE levels in ASD without diminishing the learning capabilities through synaptic formation.

When we consider the markers of systemic inflammation as markers for ASD, we may find decreased specificity because children have frequent infectious episodes, as well as subacute activation of the immune system following vaccination or during the prodromal phase of infections, or of symptomless infections, so the presence of inflammation is unlikely to be a stand-alone criterion for ASD. Adding NFkB or a more specific marker for neuroinflammation, such as GFAP, is clearly helpful, although it is likely that these two markers are elevated only in about 2/3 of ASD kids.

More specifically, alterations seen in ASD can be classified as intrinsic or extrinsic in relation to the CNS and the origination site of the possible causal factors: *intrinsic* CNS factors primarily affect the neurons, glia, astrocytes, or oligodendrocytes mostly via genetic dysfunctions (primary ASD), while *extrinsic* CNS factors primarily affect the immune system, gut microbiome, and metabolic pathways through gene vulnerabilities or exogenous factors (microorganisms and toxicants) and subsequently alter the CNS function (secondary ASD). Some of the extrinsic factors affect the CNS after modifying the blood–brain barrier (BBB); however, some factors may act simultaneously as both intrinsic and extrinsic factors, with one example being the genetic modifications affecting the glutamate receptors mGLuR. In Figure 2 below, we have attempted to summarize those factors.

Among the extrinsic CNS factors affecting its functions are those acting on the receptors involved in glucose metabolism—the peroxisome proliferator-activated receptors γ and α (PPAR-γ and PPAR-α) [68], and the receptor for advanced glycation end-products (RAGE), which, through NFkB, influences neurite outgrowth and neuronal differentiation [69,70]. PPAR-γ plays important roles in neuron differentiation and axon polarity [71].

Another extrinsic factor is the intestinal microbiome, which was shown to be different in ASD than in typically developing kids, and its modification through diets, including the Nemechek protocol (prebiotic and probiotic administration), has yielded notable results in some ASD cases. Glial over-activation towards the M1 inflammatory phenotype seems to be present in cases of intestinal dysbiosis, with simultaneous involvement of multiple mechanisms including inflammation via TNF-α, NFkB, increased GFAP and glial activation via LPS [20], alteration in the extracellular matrix via metalloproteases (MMPs), increased oxidative stress with lipid peroxidation of membranes, increased BBB permeability due to vascular and/or lymphatic damage identifiable by the increased levels of molecules such as thrombospondins (TSP-1), and vascular endothelial growth factors (VEGFs) which alter the activity of the endothelium and pericytes.

Propionic acid produced by intestinal bacteria or ingested as a food preservative is associated with brain inflammation and ASD traits [72]; moreover, fetal exposure to increased maternal lipopolysaccharides or propionic acid resulted in ASD traits in offspring with a sexual dimorphism pattern not seen with other ASD-causing agents [73].

Modifying the gut microbiota has modified levels of acetic, propionic, and butyric acids, and improved (decreased) the serotonin levels and dopamine metabolism (normalizing homovanilic acid) in ASD [74]. It was also shown that inflammatory cytokines were similarly increased in intestinal and cerebral tissues (hippocampus and amygdala), and the levels of Lactobacillus was correlated negatively with the activation of microglia, astrocytes [75], and purine metabolism in the amygdala astrocytes. Comparing urinary metabolites of ASD children with their healthy siblings showed multiple metabolic pathway perturbations involving tryptophan/serotonin, creatine, glycine, taurine, and also pantothenate and melatonin, suggesting a metabolic gut–brain link in ASD [76].

Finally, the intestinal microbiome plays an important role in the synthesis of certain neurotransmitters, such as glutamate, serotonin, and GABA [77], but butyrate—which can cross the blood–brain barrier—influences immune regulation, energy metabolism, and the transcript and translation of various genes. As a result, the gut microbiota composition was shown to correlate with neurodevelopment, and Bifidobacterium/Enterobacteriaceae vs. Bacteroides were associated with differences in toddler and infant temperament and behavior [78].

The increased levels of NSE reflecting an increase in neuronal apoptosis seem to be triggered in many cases via extrinsic pathways (ex. immune system activation with TNF-α increase) or through the activation of various metabolic or excitotoxic pathways (Figure 2). Of these, the mTOR pathway alone or simultaneously with MAPK and PPAR-γ was shown to be activated in many cases [42,79]. Simultaneous modifications of the MAPK pathway and the immune system function (IgD) were also found [53], and there are multiple possibilities of E/I modification via a disruption of distinct, multiple neurotransmitter paths: glutamatergic—mGLT via ERK and PPAR, and NMDAR via PI3K, etc.

ASD causal factors intrinsic to the CNS stem from various dysfunctions of neurons, astrocytes, or microglia, with a common occurrence being a deficit of neurotransmitter synthesis or degradation (GABA, glutamate, and NMDA), which, besides modifying the E/I balance, can also alter the mitochondria (redox status and energy production) to the point of excitotoxicity, triggering the release of caspases and apoptosis.

Another example of a causal factor is membrane alteration (lipid peroxidation, synthesis or degradation of synaptic membranes, etc.) with subsequent metabolic alterations in astrocytes and glia, and M1 glial activation. Glial cells may be over-activated by the failure of astrocytes to recover glutamate from the extracellular space, by metabolic residues triggering inflammatory-type reactions [68,71], by the modifications of glial membrane transporters, etc. Interestingly, excess glutamate (intrinsic—from astrocyte over-production or deficient metabolization—or extrinsic—from intestinal absorption or microbiome) can be lowered via the administration of either arginine or ornithine [80] (p. 398, and respective “Figure 2”) with amelioration of CNS excitotoxicity.

Astrocytes display specific purinergic-driven activities mediated via the adenosine receptors A1R and A2AR, so A1R activation favors immunosuppression, while A2AR astrocytes regulate the glutamate uptake and release via astrocytes, ATP release, the activation of Na^+^/K^+^ ATPase, and synaptic transmissions [81].

Astrocytes also play important roles in lipid and cholesterol metabolism. It was shown that sterol regulatory element binding proteins (SREBPs) are abundant in hippocampal astrocytes. Also, astrocytes control membrane integrity (including synaptic vesicles) by clearing peroxides produced from lipid degradation of membranes, and they are involved in synaptic formation and transmission [82]. In this context, lovastatin is known to inhibit ERK and downstream mTOR1 [83], and recently, administration of atorvastatin to an animal model of ASD has decreased inflammation and oxidative stress reflected in the brain levels of IL-2, IL-17, TNF-α, lactate, and malondialdehyde, as well as the increased levels of the sphingosine-1-phosphate, the nerve growth factor (NGF), and the number of neurons in the hippocampus and cerebellum [84].

Another intrinsic CNS factor is the constitutive activation of mTORC1, which is frequently a consequence of the loss-of function mutations in the phosphatase and tensin homolog (PTEN), the fragile X mental retardation protein (FMR1), and neurofibromin (NF1) and tuberous sclerosis complex (TSC1/2) genes. The lack of inhibitory modulation of these (and other) genes caused the increased activation of the phosphoinositide-dependent kinase (PDK); phosphatidylinositol 3-kinase (PI3K); and extracellular signal-regulated kinase (ERK), which in turn can be activated via the NMDA receptor (NMDAR) and also via extrinsic CNS factors via the metabotropic glutamate receptor (mGluR).

After reviewing this data, we can infer that the pervasive NSE elevation that can be seen in ASD is most likely due to the single or multiple activation of different cellular pathways, which result in apoptosis: p38/MAPK; Bcl2-Bax/caspases 3, 9; cyclines D, E; PARP/p53; CDK4 and ERK/FOS; CREB, β-catenin; pTEN/AKT; Myc, etc., where possibly the apoptotic pathways most commonly activated are those involving mTOR.

Studying mTOR inhibitors as pharmacotherapeutic intervention in neurodegenerative pathologies with ASD features (tuberous sclerosis) showed that sirolimus was safe and effective in 2-year-old children and even better tolerated in kids than adults at full-schedule dosing in other neurodegenerative pathologies linked to mTOR activation [85]. It is also possible that the lack of efficacy of mTOR inhibitors observed in some studies of tuberous sclerosis was due to the specific inhibition of the mTOR1 path, leaving mTOR2 functional; or because of the simultaneous alteration in the mTOR path with others such as Atg7 [79]; or simply because the functional alterations seen in ASD are of different intrinsic or extrinsic causalities (metabolic, inflammatory, etc.) and neuronal apoptosis is not always mTOR-driven.

The downmodulation of mTOR can be achieved with very low dose tacrolimus (1 mg/week) and with lower calcineurin activation and neuro-inflammation. Alternatively, mTOR was shown to be downmodulated via naturally occurring substances which have been used in traditional medicine for centuries, so their administration can be viewed as safe—guggulsterone decreased JAK/STAT3 activation and increased/PPAR-γ levels [86], and chrysophanol, a naturally occurring substance extracted from Rheum palmatum extract [87], which inhibits the PI3K/AKT/mTOR pathway, was shown to be over-active in ASD. Improving mitochondrial metabolism by administering succinate and/or intermittent dichloroacetic acid (DCA) as PDK (and subsequent Akt/mTOR) inhibitors, or tauroursodeoxycholic acid (TUDCA), are viewed as promising agents in ischemic brain and neurodegenerative diseases [88,89].

Even though we have found that the NSE is elevated in the vast majority of ASD kids, it is imperative that the specificity of the NSE is further studied. In addition, elevated NSE in other pathologies offers an opportunity to find specific modifications and treat these oncological pathologies via the possible differential expression of metabolic (glycolysis and HIF-1α), redox (reduced glutathione), and apoptosis pathways (Bcl/Bax, caspases, etc.) [42]. Finally, the increased NSE and neuronal destruction in ASD is an additional argument for cellular therapies, which, in some cases, can replace the affected neurons in the deficient neuronal networks and be able to function because of the different temporal circumstances [90].

## 5. Conclusions

After finding increased NSE values in the vast majority of ASD kids tested in our study, we are proposing the NSE for consideration and further evaluation as an important new biomarker for ASD which can greatly improve the diagnostic and therapeutic capabilities in this pathology. A possible direction for improvement is the specific testing of the α- and γ-NSE subunits in the plasma of ASD kids, which can determine its provenance from either neurons (γγ subunits) or microglia (αγ subunits), and direct therapeutic strategies more towards inflammation (microglia activation and the presence of the α-subunits) or metabolic/astrocyte paths (the predominance of γ-subunits). Furthermore, the specificity and usefulness of the NSE can be improved by adding pathway-specific biomarkers such as GFAP, reduced glutathione, vanilmandelic acid (VMA), homovanilic acid (HVA), and quinolinic acid (QA) using dual mass spectroscopy testing (MS/MS) of blood or urine, which can also direct treatment towards decreasing neuro-inflammation or improving the mitochondrial function and neuronal red-ox status with specific agents. Normal NSE values in a child with autistic traits but no inflammation can help guide treatment towards a genetic or growth factor deficiency.

Clinical evaluation and psychotherapy is irreplaceable as the most important diagnostic and therapeutic intervention in ASD; however, the NSE adds a material-specific component for ASD diagnosis and treatment and possibly enables earlier therapeutic interventions and better evaluation of their efficacy, allowing the child to acquire and develop improved functional neuronal networks earlier in life, with better integration in their regular educational schedule. Late therapeutic interventions in ASD are usually followed by less, more limited progress because of different developmental modifications in the CNS, which, in ASD, were shown to be age-specific [27,29,41,59].

Finally, after communicating this important finding, and as a way to overcome the limitations of the current study, we will proposeclinical studies to include testing the NSE in conjunction with one or more of the markers suggested above (GFAP, HVA, VMA, QA, glutamate, etc.), which can identify and help monitor the affected brain pathways and guide therapy in different patients with individual forms of ASD.

## Figures and Tables

**Figure 1 life-13-01736-f001:**
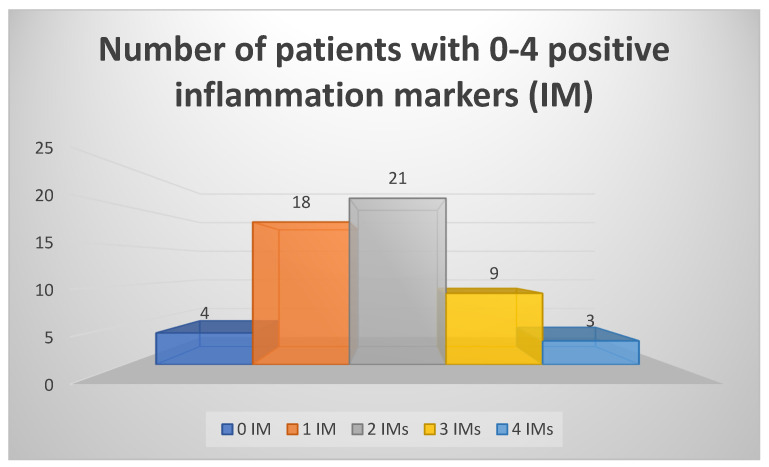
Number of inflammation markers (IM) detected simultaneously in patients.

**Figure 2 life-13-01736-f002:**
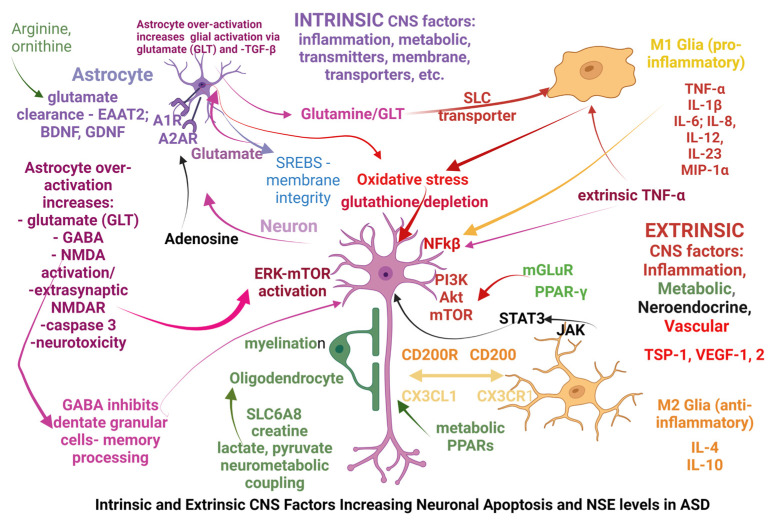
Elevated NSE levels resulting from neuronal apoptosis induced by intrinsic and extrinsic CNS factors.

**Table 1 life-13-01736-t001:** Results of inflammatory marker (IM) testing and NSE.

	ESR	γ-Globulins	Ferritin	CRP	Eosinophils	TNF-α	α-2 Globulins	NSE
Normal laboratory range	<16 mm/h	11.1–18.8%	5.3–99.9 ng/mL	<0.1 mg/dL	0–4.5%	<8.1 pg/mL	<11.8%	<16.3 ng/mL
Patients tested (n=)	55	55	51	51	55	52	55	41
Number (n=); percentage (%) of abnormal values	11; 20%	15; 27.3%	14; 27.4%	15; 29.4%	18; 32.7%	36; 69.2%	39; 70.9%	40; 97.5%
Mean ± SD	11 ± 9.9	N/A	N/A	N/A	N/A	9.22 ± 3.98	13.1 ± 4.39	24.0 ± 10.4
Number (n=) of Lower/Higher values	11 High	15 Low	10 Low/4 High	15 High	18 High	36 High	39 High	40 High

ESR—erythrocyte sedimentation rate—1 h; CRP—C-reactive protein; TNF-α—tumor necrosis factor-alpha; NSE—neuron-specific enolase.

**Table 2 life-13-01736-t002:** Other levels tested in blood: electrolyte, hematologic, metabolic, and endocrine.

	Low MCV, Low MHC	High RBC, Low MCV (Thalassemia)	Low Na (<138 mmol/L)	High MCV	Urea	Liver Function Tests	Homocysteine	IGF-1; hGH
Number, percentage	1/55; (1.81%)	2/55; (3.63%)	4/55; (7.27%)	5/55; (9.09%)	5/55; (9.09%)	6/55; (10.9%)	5/30; (16.6%)	2/10; (20%)
Mean ± SD	Low/Low	High/Low	Low	High	High	High	4 Low, 1 High	Low

RBC—red blood cells; MCV—mean corpuscular volume of RBCs; MHC—mean hemoglobin concentration; Na—sodium; IGF-1—insulin like growth factor-1; hGH—human growth hormone); d-direction of modification.

## Data Availability

Due to privacy concerns, individual testing data are not publicly posted; however, the global data tabulated in Excel format is available upon request and the signing of a confidentiality agreement. Please send written request to felicians@bio-forum.net.

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
