# Peer review of "Neuron-Specific Enolase (NSE) as a Biomarker for Autistic Spectrum Disease (ASD)"

_life, 2023, doi:10.3390/life13081736_

Round 1
Reviewer 1 Report
Based on the title of the submitted paper, it was expected that more specifics would be presented, including robust experimental results for NSE as a potential biomarker for autism.
Rather than simply describing the various factors of autism, it is thought that more detailed evidence, explanations, and discussions on the potential factors of autism biomarkers so far are needed.
Whether simple increases in NSE values observed in plasma and serum parameters of autistic patients can serve as biomarkers for autistic patients and the statistical significance of the results are also questionable. In addition, the possibility of NSE peptide as a biomarker of autism already has been mentioned in a previous report (Peptides of neuron specific enolase as potential ASD biomarkers: From discovery to epitope mapping. Brain Behav Immun. 2020 Feb;84;200-208. doi: 10.1016/j.bbi.2019.12.002. Epub 2019 Dec 5.), more specific and robust content is expected in this paper.
Overall, more specific evidence and discussion of NSE as a biomarker of autism is needed including the presentation and analysis of convincing results.
Also, as the author mentioned in the last part, it would be a more persuasive paper if the possibility of NSE as a marker of autism is presented in more detail in comparison with other possible biomarkers.
Minor revisions or rephrase, if applicable would be recommended.
Author Response
Please see the attachment, Thank you!

Reviewer 2 Report
In this study, the authors report NSE as a new biomarker for ASD. The data of this study are interesting; however, the following minor comments should be addressed.
1) Introduction: The rationale of the measurement of NSE should be stated in the introduction.
2) Are there any correlations between blood levels of NSE and severity of ASD symptoms?
3) Are there any correlations between blood levels of NSE and other biomarkers in ASD patients?
4) The authors discussed role of gut microbiota in the discussion section although they did not examine the gut microbiota composition in ASD patients. Gut microbiota dysbiosis can contribute to systemic inflammation. Please discuss about systemic inflammation by dysbiosis of gut microbiota.
Minor English editing is needed.
Author Response
Please see the attachment, thank you!

Round 2
Reviewer 1 Report
Though additional corrections could hardly be found in the revised version, the author's rebuttal was convincingly presented and most of them were understandable. It is still somewhat regrettable that statistically significant quantitative data could not be included in this submitted paper. However, according to the author's perspective and plan, it is believed that corresponding data will be presented in the next prepared paper as the authors mentioned.
Additionally, it is recommended to add the following.
1. It would be better if you briefly re-state the summary of the research results and the significance of the research results in the conclusion section.
2. Any related evidence regarding NSE as a possible marker for ASD including the aforementioned paper (Peptides of neuron specific enolase as potential ASD biomarkers: From discovery to epitope mapping. Brain Behav Immun. 2020 Feb;84;200-208. doi: 10.1016/j.bbi.2019.12.002. Epub 2019 Dec 5.) might be beneficial to be added to the references, and it would be informative if you briefly emphasize the difference from the corresponding paper in the discussion section.
Author Response
Dear Editor,
I have attached the reply to the new observations of Reviewer 1 - please see attached, and also I have made a few modifications in the manuscript, by deleting and adding text and references
I will attach the new revised manuscript as indicated.
Best regards,
Dr Felician Stancioiu
